# Determinants of undernutrition and overnutrition among reproductive-age women in Bangladesh: Trend analysis using spatial modeling

**Richa Vatsa[1], Umesh Ghimire[2,3]\*, Khaleda Yasmin[4], Farhana Jesmine Hasan[5]**

**1** Central University of South Bihar, Gaya, Bihar, India, **2** Richard M. Fairbanks School of Public Health, Indiana University, Indianapolis, Indiana, United States of America, **3** School of Public Health, University of Minnesota, Minneapolis, Minnesota, United States of America, **4** Family Planning-Field Services Delivery, Directorate General of Family Planning, DGFP, Dhaka, Bangladesh, **5** Initiatives for Married Adolescent Girl's Empowerment (IMAGE) Project, Dhaka, Bangladesh

\* umghimi@iu.edu

**Data Availability Statement:** The data used in this study can be accessed without any cost by registering at https://dhsprogram.com/Data/.

## Abstract

### Background

Bangladesh is facing a dual burden of malnutrition, with high rates of undernutrition and increasing rates of overnutrition. The complex scenario of malnutrition in Bangladesh varies across different regions, making it a challenging public health concern to address.

### Objectives

This study analyzes the spatial and temporal dependence of underweight and overweight Bangladeshi women of reproductive age.

### Methods

Nationally representative cross-sectional data from the Bangladesh Demographic and Health Surveys in 2014 and 2017–18 were utilized to study the changes in weight status in 15–49-year-old women who were either underweight or overweight. A Bayesian geo-additive regression model was used to account for non-linear and linear effects of continuous and categorical covariates and to incorporate spatial effects of geographical divisions.

### Results

The prevalence of overweight or obese women in rural, city corporations, and other urban areas increased significantly over the four years from 2014 to 2017–18. Women in the categories 'richer' and 'richest' were more likely to be overweight or obese. Women from Sylhet were more likely to be underweight in both survey years; however, the spatial effects were significant for underweight women in Mymensingh for the year 2017–18. Women in Rajshahi and Khulna were more likely to be overweight or obese in 2014, and women from Barishal and Chittagong were more likely to be overweight in the year 2017–18.

**Funding:** The authors received no specific funding for this work.

**Competing interests:** The authors declare that they have no competing interests.

## Conclusions

Underweight and overweight statuses in women vary unevenly across Bangladesh, with a substantially higher prevalence of overweight or obese women in more urbanized areas. The growing burden of overweight and obesity among Bangladeshi women should be addressed with interventions aimed at those in the reproductive age group.

## Introduction

Malnutrition in various forms is recognized as an emerging and serious public health concern, resulting in increased healthcare costs, decreased productivity, and reduced quality of life in both low-middle-income countries (LMICs) and developed countries [1, 2]. Undernutrition and obesity are disproportionately prevalent in LMICs, posing complex and multi-faceted challenges in dealing with the global complexities of nutritional problems [3–6]. By 2030, the Sustainable Development Goal (SDG) 2, "Zero Hunger," Target 2.1 and 2.2, emphasizes achieving food security and improved nutrition and promoting sustainable agriculture, and eradicating all forms of malnutrition [7]. Numerous factors influence nutritional intake, including insufficient dietary energy intake, poor absorption, and biological use of nutrients resulting from repeated illnesses [8, 9]. Nevertheless, the non-availability of quality diets compounds a higher risk of undernutrition and overweightness. Moreover, sedentary lifestyle, junk food high in sugar and unsaturated fats, and a lack of exercise are all recognized risk factors for becoming overweight [4, 10]. In recent decades, demographic and socioeconomic shifts have contributed to a sharp increase in the trend of overweight and obese people in LMICs [5, 11]. Due to these risk factors, the exponential rise in non-communicable diseases (NCDs) places South Asian countries under a double burden of malnutrition [12, 13].

Bangladesh is divided administratively into eight major divisions and sixty-four districts. Significant differences exist between the 22% of Bangladeshis who live in urban regions and the 52% of rural residents, with 32% of Bangladeshis living below the poverty line of 1.25 US dollars a day. Wealth inequality among poor and rich populations is extensive, with the poorest 20% of the population possessing nine percent of the wealth and primarily residing in cities [14]. Bangladesh, like many other developing countries, has experienced major demographic and epidemiological shifts over the last two decades [15]. Bangladesh's population is expected to exceed 202 million by 2050, a significant increase from 147 million in 2007 [16]. Rapid urbanization due to rapid population growth and migration has caused a shift in the population's dietary habits. The coexistence of undernutrition and obesity can be catastrophic to the country's already frail healthcare system. A growing body of evidence indicates a rising prevalence of both underweight and overweight among Bangladeshi women [17, 18]. The National Plan of Action for Nutrition (2016–2025) has envisioned reducing underweight (BMI<18.5 kg/m$^2$) adolescents to below 15% and decreasing overweight mothers (BMI>23 kg/m$^2$) to 30% [19]. However, the trend toward being overweight is 2.6 times higher than the last survey conducted a decade ago. According to the latest Bangladesh Demographic and Health Survey (2017–18), malnutrition among women in Bangladesh was 12%, and the prevalence of overweight and obese women was 32% [20].

Earlier studies estimated the prevalence and risk factors of BMI while ignoring several unobserved covariates that could influence the association. Our study addresses the limitations of previous studies by accounting for the spatial dependence of underweight and overweight women using a Bayesian geo-additive regression method [21, 22]. One of the reasons for

utilizing the Bayesian geo-additive model rather than the traditional linear regression in the study is its flexibility to incorporate the spatial and non-linear effects in the model. As a result, this method is appropriate for outlining a broader effect of multiple outcomes and can identify the effects of the covariates in the model.

The results found in this paper should be a useful starting point for the National Nutrition Plan to formulate policy and further research based upon the current trends and risk factors affecting malnourishment and overweightness in women. The use of geospatial analysis is useful in estimating and assessing the nutritional status of reproductive-age women across several divisions in Bangladesh.

## Methods

### Data source and study design

This study relied on two consecutive nationally representative Bangladesh Demographic and Health Surveys (BDHS) of 2014 and 2017–18 (available at www.dhsprogram.com). In collaboration with the National Institute of Population Research and Training in Bangladesh and the United States Agency for International Development, BDHS collects data on several aspects of socio-demographic, fertility, family planning, maternal and child health, and other health and nutrition indicators using a set of standard questionnaires. The BDHS samples were collected from households using a two-stage stratified cluster sampling method. The Primary Sampling Units (PSUs) for the 2014 BDHS were determined using the 2011 national population and housing census. In the BDHS of 2014, a total of 17,500 households were interviewed, with a response rate of 98 percent. In contrast, the BHDS of 2017–18 included 19,457 households, with a 96.5 percent response rate. More information on the survey and its methodology can be found in the BDHS reports [20, 23]. The National Research Ethics Committee approved the Bangladesh Demographic and Health Survey. Data collection procedures were approved by the ORC Macro (Macro International Inc) Institutional Review Board. Written informed consent was obtained from all participants for inclusion in the BDHS.

The two major inclusion group criteria in the analysis were (1) non-pregnant, never-married women aged 15 to 49 years and (2) women who had given birth two months before the survey. After considering these criteria and eliminating 35 missing cases for 'reading newspaper or magazine' (24), 'listening to radio' (1), 'watching television' (1), 'respondent currently working' (5), 'source of drinking water' (3), and 'type of toilet facility' (1), a total of 14,997 women from the BDHS 2014 survey were included in the analysis. Furthermore, 16,751 women from the BDHS 2017–18 survey were considered in the analysis using the aforementioned inclusion criteria and removing one missing case for listening to radio.

### Dependent variable

Body Mass Index (BMI) is calculated using the formula weight (in kilograms) divided by height (in meters) squared. According to WHO Expert Consultation (2004) for Asian-specific BMI classification, underweight was defined as having a BMI less than 18.5 kg/m$^2$, normal as having a BMI of 18.5 to 23.0 kg/m$^2$, and overweight as having a BMI of 23.0 to 27.5 kg/m$^2$. Obesity was defined as having a BMI greater than 27.5 kg/ m$^2$ [6].

### Independent variables

The women were asked to complete standard DHS questionnaires on a variety of sociodemographic variables such as place of residence, household wealth status on a five-point scale, education of respondents (divided into five categories: no education, primary, secondary, higher,

and more), and work status. Household characteristics included in the study were toilet facility (improved/unimproved), water source (protected/unprotected) characterized per WHO definitions, availability of electricity in house (yes/no or not a de jure resident) [24]. Women's exposure to mass media such as reading newspapers/magazines, listening to the radio, or watching television, was reported as 'yes' for 'at least once a week' and 'almost every day', and 'no' for 'not at all' and 'less than once a week'. For spatial analysis, administrative divisions were used, and the continuous variable 'age of women' was considered for possible nonlinear effects on underweight and overweightness among women.

## Statistical analysis

In the equation below, let $y_i$ denote the nutritional status of $i^{th}$ women, with possible categories as 'underweight', 'normal weight', 'overweight/obese', and, $x_i$ be the vector of covariates that $y_i$ may relate to via a multinomial logit model, defined as,

$$P[y_i = r] = \frac{\exp(\eta_{i,r})}{1 + \sum_{s=1}^{K-1} \exp(\eta_{i,s})}, r = 1 : (K-1).$$

The term $r$ stands for a category of nutritional status; $K$ is the total number of categories.

Six multinomial logit models based on various functional definitions of the predictor, $\eta_{i,r}$ were considered for modelling the nutritional status under study. The models are defined as follows:

$$\text{M1}: \ \eta_{i,r} = u_i'\beta_r \ r = 1 : (K-1),$$

$$\text{M2}: \ \eta_{i,r} = u_i'\beta_r + f_r(z_i), \ r = 1 : (K-1),$$

$$\text{M3}: \ \eta_{i,r} = u_i'\beta_r + g_r(S_i) + h_r(S_i), \ r = 1 : (K-1),$$

$$\text{M4}: \ \eta_{i,r} = u_i'\beta_r + f_r(z_i) + g_r(S_i), \ r = 1 : (K-1),$$

$$\text{M5}: \ \eta_{i,r} = u_i'\beta_r + f_r(z_i) + h_r(S_i), \ r = 1 : (K-1),$$

$$\text{M6}: \ \eta_{i,r} = u_i'\beta_r + f_r(z_i) + g_r(S_i) + h_r(S_i), \ r = 1 : (K-1),$$

The term $u_i$, a vector of categorical covariates, $z_i$, age, and $S_i$, refers to discrete indices representing geographical divisions. The term $\beta_r$ relates to linear effects, the term $f_r$, a nonlinear effect of age, whereas $g_r$ and $h_r$ stand for structured and unstructured spatial effects of divisions, respectively.

The structured spatial effects were included to account for variation in data due to latent factors as a result of the spatial autocorrelations between neighboring regions. These unstructured spatial effects by region explain the spatial variation in data due to local unobserved factors. Further, adding spatial effects in the model to account for spatial autocorrelation reduces the bias in estimation by explaining the extra variation in the data due to spatial factors.

The model M1 considers the linear effects of all the categorical and continuous explanatory variables, and model M2 includes both linear and nonlinear effects of categorical and continuous predictors, respectively. On the other hand, model M3 comprises linear effects of all explanatory variables and the structured and unstructured spatial effects of divisions. Model M4 and M5 both include linear and nonlinear effects, but they differ in inclusion of spatial effects. Model M4 includes structured spatial effects, whereas model M5 considers

unstructured spatial impacts. The final model M6 comprises all linear, nonlinear, and structured and unstructured spatial effects.

The above mentioned geo-additive regression models contain modifications to GLMs which account for non-linear effects of continuous covariate, 'age', spatial effects of geographical divisions of the country, and linear effects of other categorical covariates on the same platform [25]. Moreover, geo-additive models are preferred over the linear regression models as the later accounts for linear effects only, whereas, the geo additive models allow for the inclusion of possible nonlinear impacts of predictors. Geo-additive models also include the structured or unstructured spatial effects which may explain spatial variation in data due to unobserved factors associated with spatial locations and spatial autocorrelation (effect due to dependence or similarity between neighboring regions).

In this study, we followed the Bayesian approach to inference to fit the model. The linear effects $\beta_r$ were assigned diffuse priors, $P(\beta_r) \propto const$. The Bayesian p-spline priors were assumed over the unknown nonlinear smooth function $f_r$ [21]. The nonlinear function $f_r(z_i)$ was approximated as a linear combination of $J = (m+l)$ B-spline basis functions, $B_j(.)$, using a penalized polynomial spline of degree $l$. Symbolically,

$f_r(z_i) = \sum_{j=1}^{J} \vartheta_j B_j(z_i), r = 1 : 2$, and $z_{i,min} = \zeta_{i,0} < \zeta_{i,1} < \cdots < \zeta_{i,m} = z_{i,max}$. That is, the space of $Z_i$ was defined by equally spaced knots. The accuracy of this approximation can be maintained with the consideration of a large number of knots. In Bayesian settings to this spline approximation, the unknown parameters $\vartheta_j$ were assumed with second-order random walk priors, i.e.,

$$\vartheta_j = 2\vartheta_{j-1} - \vartheta_{j-2} + e_j, \text{with} e_j \sim N(0, \sigma_e^2).$$

An inverse-gamma prior density with pre-specified hyper-parameters $\omega_1$ and $\omega_2$ was considered for the variance parameter $\sigma_e^2$. To note, small estimates of variance parameter $\sigma_e^2$ shall ensure smooth estimation to the nonlinear effect $f_r(z_i)$, and vice-versa.

Additionally, a Gaussian Markov random field prior density [26] was assumed for the unknown structured spatial effects $g_r(S_i)$. In GMRF prior densities, the neighboring spatial locations are assumed correlated to account for their similar environmental, social, or cultural settings. Further, the adjacent spatial locations sharing common boundaries are treated as neighbors and therefore, are considered correlated. The conditional distribution of latent spatial effects $g(S_{i,k})$ of the $k^{th}$ spatial location $S_{i,k}$ at which the $i^{th}$ woman resides, for given spatial effects $g(S_{i,-k})$ of other spatial locations $(S_{i,-k})$, is defined as

$$g\left(S_{i,k}\right)|g\left(S_{i,-k}\right) \sim N\left(\frac{1}{|N(k)|}\sum_{l \in N(k)} g(S_{i,\,l}), \; \frac{\delta}{|N(k)|}\right), \; \delta \sim inverse-gamma(a, b).$$

The term $N(k)$ refers to the vector of neighbors to the $k^{th}$ spatial location $S_{i,k}$. The mean parameter of the above GMRF prior is considered as the average of neighbors of the $k^{th}$ spatial location and the variance parameter is taken as inversely proportional to $|N(k)|$ [26]. A non-informative inverse-gamma hyper-prior density (with hyper-parameters a = b = 0.001 as the default setting) was considered over the variance parameter $\delta$ of the GMRF prior.

The unstructured spatial effects $h_r(S_i)$, were assumed to be independently and identically distributed with a Gaussian density a Priori with an unknown variance parameter. A non-informative inverse-gamma prior density with hyper-parameters 0.001, and 0.001, respectively, was also considered for the variance parameter.

Variance parameters of spatial effects provide a trade-off between model complexity and smoothness in estimation of spatial effects. Thus, small values of the variance parameters are

preferred to achieve better estimates of spatial impacts. More details on the above geo-additive regression model for categorical response variables may be found elsewhere [27].

The reason for considering the six models under study is to choose the best model to fit the data. The deviance information criterion (DIC) was considered to select the best model among the six models the model with the least DIC is considered the best [28]. The models with differences in DIC values equal to or less than five may be taken as similar in fitting data. DIC is defined as follows: $DIC = \bar{D} + p_D$, the term $\bar{D}$ is the posterior mean of model deviance, used as a measure of goodness of fit. Whereas, the effective number of parameters, $p_D$ refers to a measure of model complexity and guards against overfitting of data.

Further, the complexity of Bayesian computation involved in the inference procedure was addressed with the application of MCMC techniques via software BayesX using R-package BayesXSrc designed for the Bayesian inferences in geo-additive models [29]. A total of 35000 MCMC iterations with 5000 burn-in period, 10 steps, with default hyperparameter setting $a = b = c = d = 0.001$, and default degree of spline 3 with 20 equidistant knots in the spline was considered for the convergence of MCMC samples for Bayesian inference in each of the above-mentioned six model settings.

## Results

Table 1 presents the distribution and trends of background characteristics of reproductive-age women along with p-values indicating the statistical significance of the change in percentage over the two-survey period. The t-test for proportion was utilized to compute the p-values using appropriate R codes. For both surveys, the majority of women lived in rural areas, followed by urban areas. The percentage of women residing in urban increased from 22.56% in 2014 to 26.43% in 2017–18. For both surveys, nearly equal percentages of women (19%) belonged to the 'poorest' and 'poorer' wealth quintiles, while around 21% belonged to the 'richer' and 'richest' quintiles. During the two-survey period, women's educational status improved at all educational levels. Similarly, the percentage of working women increased significantly from 33.65% in 2014 to 51% in 2017–18.

The percentage of women who had access to safe drinking water decreased slightly in 2017–18 from 2014. The results show that access to improved toilet facilities decreased by four percentage point during the two surveys. Access to electricity, on the other hand, increased by more than 15%. The percentage of women who read newspapers and listened to the radio also decreased. In comparison to the 2014 survey, three percent more women watched television at least once a week. The percentage of women being underweight declined significantly but there was a 10% increase in women with overweight/obesity. Except for the wealth categories (poorer, middle, richer, and richest), the changes in the percentages of characteristics over the two-survey period were significant as indicated by the p-values in Table 1.

Table 2 shows the results for DIC, deviance values, etc. for each of the six models fitted for both surveys- 2014 and 2017–18. As per the DIC results, models M4, M5, and M6 have the least DICs for both surveys. The differences of these DIC values are less than five, indicating that any of the three models can be considered equally good for model fitting of the data. Therefore, model M4 which accounts for linear, nonlinear, and structured spatial effects, was selected for estimation of these effects on data.

### Linear effects

Table 3 presents the findings of linear association of underweight and overweight/obesity (referenced to the normal weight) with categorical background characteristics of reproductive-age women in Bangladesh. Although, the differences were not statistically significant, women in

**Table 1. Frequency distribution of selected characteristics of reproductive age-group women in Bangladesh.**

| Variable | BHDS 2014 | | BHDS 2017–18 | | p-value[#] |
|---|---|---|---|---|---|
| | N | (%) | N | (%) | |
| **Residence** | | | | | |
| City Corporation | 1782 | 11.88 | 1630 | 9.73 | <0.001*** |
| Other Urban | 3383 | 22.56 | 4428 | 26.43 | <0.001*** |
| Rural | 9832 | 65.56 | 10693 | 63.84 | 0.001*** |
| **Wealth quintile** | | | | | |
| Poorest | 2774 | 18.5 | 3249 | 19.4 | 0.041** |
| Poorer | 2841 | 18.94 | 3252 | 19.41 | 0.288 |
| Middle | 3027 | 20.18 | 3273 | 19.53 | 0.150 |
| Richer | 3148 | 21.0 | 3378 | 20.17 | 0.069 |
| Richest | 3207 | 21.38 | 3599 | 21.49 | 0.827 |
| **Educational level** | | | | | |
| No Education | 3884 | 25.9 | 2980 | 17.79 | <0.001*** |
| Primary | 4521 | 30.15 | 5579 | 33.31 | <0.001*** |
| Secondary | 5356 | 35.71 | 6258 | 37.36 | 0.002*** |
| Higher | 1236 | 8.24 | 1934 | 11.54 | <0.001*** |
| **Working Status** | | | | | |
| No | 9951 | 66.35 | 8185 | 48.86 | <0.001*** |
| Yes | 5046 | 33.65 | 8566 | 51.14 | <0.001*** |
| **Water Source** | | | | | |
| Protected | 13942 | 92.97 | 15378 | 91.8 | <0.001*** |
| Unprotected | 1055 | 7.03 | 1373 | 8.2 | <0.001*** |
| **Toilet Facility** | | | | | |
| Improved | 10405 | 69.38 | 10963 | 65.45 | <0.001*** |
| Unimproved | 4592 | 30.62 | 5788 | 34.55 | <0.001*** |
| **Electricity** | | | | | |
| No or Not a Dejure Resident | 5753 | 38.36 | 3846 | 22.96 | <0.001*** |
| Yes | 9244 | 61.64 | 12905 | 77.04 | <0.001*** |
| **Reads Newspaper** | | | | | |
| No | 14078 | 93.87 | 16152 | 96.42 | <0.001*** |
| yes | 919 | 6.13 | 599 | 3.58 | <0.001*** |
| **Listens to Radio** | | | | | |
| No | 14663 | 97.77 | 16465 | 98.29 | <0.001*** |
| yes | 334 | 2.23 | 286 | 1.71 | <0.001*** |
| **Watches Television** | | | | | |
| No | 7344 | 48.97 | 7692 | 45.92 | <0.001*** |
| yes | 7653 | 51.04 | 9059 | 54.08 | <0.001*** |
| **Nutritional Status** | | | | | |
| Underweight | 2730 | 18.2 | 1962 | 11.71 | <0.001*** |
| Normal | 6155 | 41.04 | 6294 | 37.58 | <0.001*** |
| Overweight/obese | 6112 | 40.76 | 8495 | 50.71 | <0.001*** |

**Significant at level of significance ≤0.05

*** significant at level of significance ≤ 0.01

# calculated using t-test

**Table 2. Comparison of models based on deviance information criterion (DIC).**

| Survey year 2014 | Model | Deviance | pD | DIC | Difference in DICs |
|---|---|---|---|---|---|
| | M1 | 28973.68 | 33.7191 | 29007.4 | 959.4 |
| | M2 | 28193.67 | 43.7292 | 28237.4 | 189.4 |
| | M3 | 28771.02 | 44.8767 | 28815.9 | 767.9 |
| | M4 (a) | 27992.11 | 55.1918 | 28047.3 | -0.7 |
| | M5 (b) | 27992.38 | 55.3228 | 28047.7 | -0.3 |
| | M6 (c) | 27992.38 | 55.6164 | 28048.0 | Reference |
| Survey year 2014 | Model | Deviance | pD | DIC | Difference in DICs |
| | M1 | 30424.7 | 33.7041 | 30458.4 | 950.7 |
| | M2 | 29724.79 | 43.1069 | 29767.9 | 260.2 |
| | M3 | 30141.09 | 46.6112 | 30187.7 | 680 |
| | M4 (a) | 29451.12 | 56.5782 | 29507.7 | 0 |
| | M5 (b) | 29450.79 | 56.1082 | 29506.9 | -0.8 |
| | M6 (c) | 29451.09 | 56.6133 | 29507.7 | Reference |

cities and other urban areas were more likely to be overweight/obese than women in rural municipalities.

According to both surveys, poorer women were more likely to be underweight and less likely to be overweight/obese. Simultaneously, middle-class women were found to be significantly less likely to be overweight/obese in the BDHS 2014 survey. Women in the richer and richest wealth categories were more likely to be overweight/obese and less likely to be underweight in both surveys; however, in the BDHS 2017–18 survey, richer wealth category had insignificant effect on the likelihood of women being underweight.

The results of the 2014 and 2017–18 surveys in terms of women's education and employment status were consistent. Women with primary and secondary education were less likely to be underweight and more likely to be overweight/obese. Higher educated women were less likely to be overweight/obese, but more likely to be underweight. Likewise, currently working women were significantly less likely to be overweight or obese. Reading newspapers was associated with women being less likely to be underweight in 2014; however, they were more likely to be overweight/obese in both surveys. Similarly, women who watched television were more likely to be overweight or obese, according to the two surveys. While access to water, toilet facilities, and electricity, as well as listening to the radio, were not significantly associated with any form of malnutrition in women.

## Spatial effects

The posterior mapping of different forms of malnutrition among reproductive-age women in Bangladesh are depicted in Figs 1 and 2. According to Fig 1A and 1b, in 2014, women from the Sylhet (7) were more likely to be underweight, whereas women from the Khulna (4) were less likely to be underweight. Women in Mymensingh (5) (previously Dhaka (3)) and Sylhet (8) were more likely to be underweight in 2017–18 (Fig 2A and 2B). In the same year, women in Barishal (1) and Chittagong (2) were significantly less likely to be underweight.

In contexts of overweight/obesity, women in the Rajshahi (5) and Khulna (4) regions were more likely to be overweight or obese in 2014. Sylhet (7) women, on the other hand, were less likely to be overweight or obese (Fig 1C and 1D). However, in 2017–18, women in Khulna (4), Barishal (1), and Chittagong (2) were more likely to be overweight or obese. Similarly, overweight/obesity was rare in the Sylhet (8) region and Mymensingh (5), which was previously included in Dhaka (3) (Fig 2C and 2D). In contrast, the results for BMI in the other divisions were not significant.

**Table 3. Posterior estimates for the linear effects of underweight, overweight, and obesity of women of reproductive age-group (15 to 49 years).**

| Variables | BHDS 2014 | | | | BHDS 2017–18 | | | |
|---|---|---|---|---|---|---|---|---|
| | Underweight | | Overweight/obesity | | Underweight | | Overweight/obesity | |
| | Mean | 95% CI | Mean | 95% CI | Mean | 95% CI | Mean | 95% CI |
| **Rural Municipality (ref)** | | | | | | | | |
| City Corporation | 0.106 | (-0.026, 0.242) | 0.15* | (0.062, 0.241) | -0.10 | (-0.278, 0.075) | 0.038 | (-0.053, 0.129) |
| Other Urban | -0.075 | (-0.17, 0.024) | 0.033 | (-0.035, 0.103) | 0.103 | (-0.009, 0.213) | 0.036 | (-0.027, 0.098) |
| **Wealth Index** | | | | | | | | |
| Poorest (ref) | | | | | | | | |
| Poorer | 0.221* | (0.123, 0.323) | -0.246* | (-0.334, -0.158) | 0.192* | (0.081, 0.302) | -0.332* | (-0.403, -0.258) |
| Middle | 0.003 | (-0.096, 0.1) | -0.106* | (-0.183, -0.029) | 0.023 | (-0.089, 0.139) | -0.033 | (-0.106, 0.038) |
| Richer | -0.193* | (-0.303, -0.082) | 0.177* | (0.097, 0.258) | -0.108 | (-0.228, 0.016) | 0.118* | (0.046, 0.193) |
| Richest | -0.389* | (-0.549, -0.228) | 0.64* | (0.536, 0.747) | -0.454* | (-0.637, -0.267) | 0.642* | (0.55, 0.734) |
| **Education** | | | | | | | | |
| No education (ref) | | | | | | | | |
| Primary | -0.269* | (-0.414, -0.121) | 0.464* | (0.346, 0.583) | -0.21* | (-0.378, -0.049) | 0.477* | (0.363, 0.588) |
| Secondary | -0.303* | (-0.586, -0.032) | 0.63* | (0.441, 0.817) | -0.271 | (-0.532, 0.003) | 0.508* | (0.349, 0.661) |
| Higher | 0.713* | (0.292, 1.143) | -1.339* | (-1.67, -1.003) | 0.514* | (0.038, 0.982) | -1.311* | (-1.624, -0.987) |
| **Currently working** | | | | | | | | |
| No (ref) | | | | | | | | |
| Yes | -0.027 | (-0.076, 0.022) | -0.127* | (-0.168, -0.085) | 0.013 | (-0.043, 0.068) | -0.12* | (-0.157, -0.084) |
| **Water sources** | | | | | | | | |
| Unprotected (ref) | | | | | | | | |
| Protected | -0.026 | (-0.115, 0.066) | 0.042 | (-0.046, 0.132) | 0.047 | (-0.062, 0.154) | -0.013 | (-0.088, 0.072) |
| **Toilet facilities** | | | | | | | | |
| Unimproved (ref) | | | | | | | | |
| Improved | -0.049 | (-0.101, 0.006) | 0.016 | (-0.036, 0.067) | -0.053 | (-0.114, 0.009) | 0.032 | (-0.015, 0.075) |
| **Electricity** | | | | | | | | |
| No (ref) | | | | | | | | |
| Yes | -0.027 | (-0.094, 0.04) | 0.023 | (-0.039, 0.082) | 0.007 | (-0.07, 0.077) | 0.019 | (-0.04, 0.076) |
| **Reading newspaper** | | | | | | | | |
| No (ref) | | | | | | | | |
| Yes | -0.172* | (-0.351, -0.006) | 0.181* | (0.089, 0.277) | 0.151 | (-0.085, 0.377) | 0.213* | (0.097, 0.33) |
| **Radio listening** | | | | | | | | |
| No (ref) | | | | | | | | |
| Yes | -0.037 | (-0.208, 0.122) | -0.11 | (-0.235, 0.02) | -0.05 | (-0.283, 0.174) | -0.124 | (-0.259, 0.008) |
| **Television watching** | | | | | | | | |
| No (ref) | | | | | | | | |
| Yes | -0.054 | (-0.114, 0.008) | 0.137* | (0.088, 0.184) | -0.044 | (-0.105, 0.018) | 0.124* | (0.084, 0.165) |
| Variance parameter of nonlinear effects of age covariate | 0.004 | (0.0008, 0.0147) | 0.005 | (0.001, 0.0184) | 0.003 | (0.0005, 0.0109) | 0.004 | (0.0010, 0.0141) |
| Variance parameter of spatial effects of divisions | 0.167 | (0.037, 0.568) | 0.149 | (0.033, 0.515) | 0.260 | (0.072, 0.784) | 0.196 | (0.054, 0.597) |

*Significant mean value as zero value is included in the 95% credible interval; Reference category is the Normal weight

Ref: reference categories

Estimates for the variance parameter of the spatial effects of division for both surveys are presented in Table 3. As per the results, small posterior mean values of the variance parameters indicate good approximations to the spatial effects for both surveys.

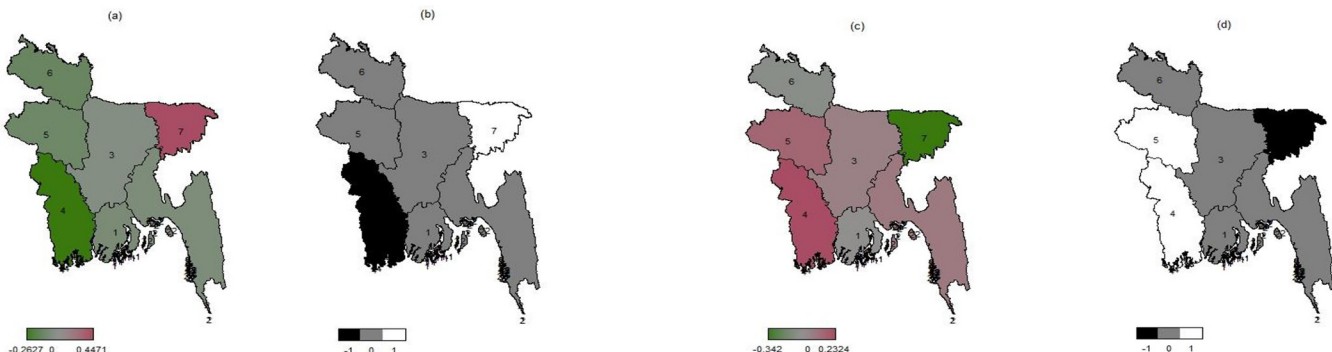

**Fig 1.** Division maps of Bangladesh showing spatial effects of: a) underweight and c) its 95% CI; b) overweight/obesity and d) its 95% CI; among reproductive age-group women BDHS 2014.

Fig 3 shows the distribution of women's BMI by place of residence for the survey years 2014 and 2017–18. Over the four years from 2014 to 2017–18, the prevalence of underweight in rural, city corporation, and other urban areas decreased. During the same time, however, the rates of overweight/obesity increased in all three residences.

## Non-linear effects

Figs 4 & 5 illustrate the non-linear effects of women's ages on nutritional categories using smooth curves. According to Figs 4A and 5A, for both survey years, women were found more likely to be underweight up to the age of 23 years. However, they were less likely to be underweight after the age of 27.

Fig 4B (for the year 2014) shows a steady increase in the likelihood of overweight/obesity around the age of 35, followed by a slight decline. Nevertheless, for the survey 2017–18, odds of women being overweight/obesity started rising after the age of 35 and peaked around the age of 40, implying that as women aged from 35 to 40, they became more overweight/obese (Fig 5B). However, a slow decrement beyond age 40 was observed in the likelihood of women to be overweight/obese. Furthermore, both surveys' results show that women up to the age of 27 were consistently less likely to be overweight or obese.

Estimates for the variance parameter of the nonlinear effects of age covariate for both surveys are presented in Table 3; the posterior mean values are small indicating a good estimate of the nonlinear effects for both surveys.

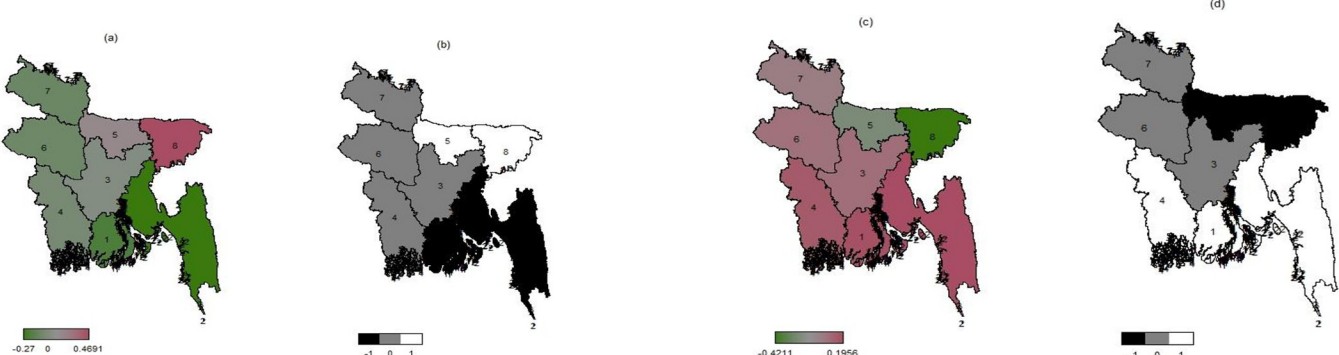

**Fig 2.** Division maps of Bangladesh showing spatial effects of: a) underweight and c) its 95% CI; b) overweight/obesity and d) its 95% CI; among reproductive age-group women BDHS 2017–18.

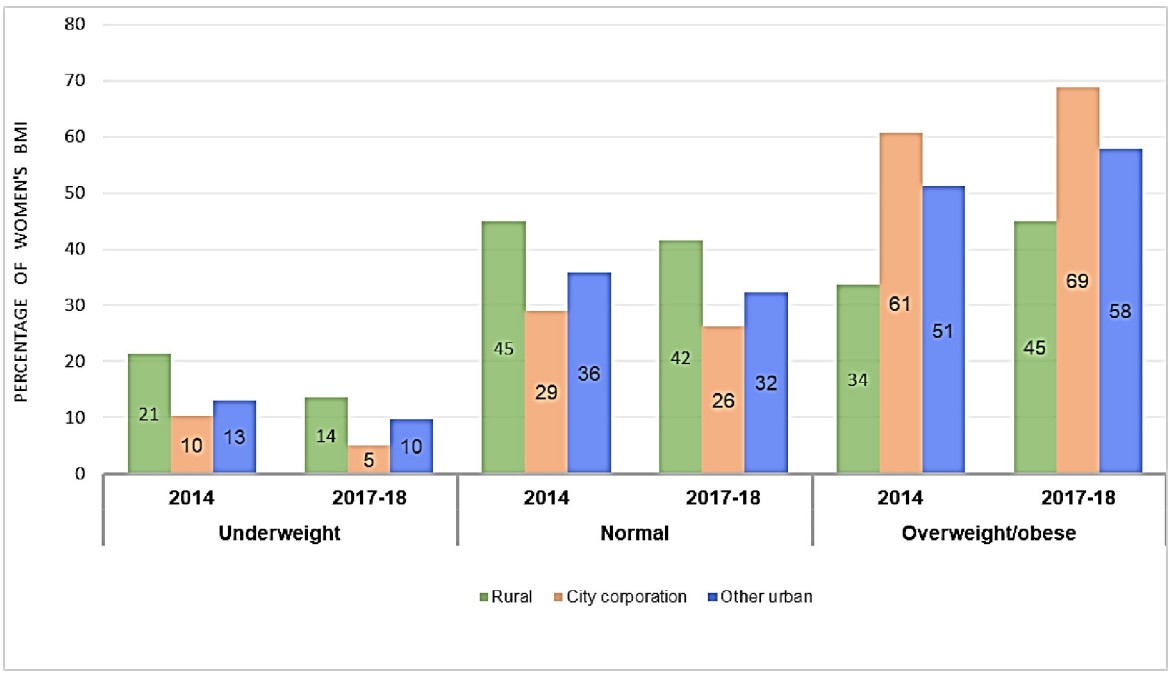

**Fig 3. Distribution of underweight, normal weight and overweight/obese according to the place of residence of 15 to 49 years women for the year 2014 and 2017–18 in Bangladesh.**

## Discussion

The present study examines the determinants of various forms of malnutrition among reproductive-age Bangladeshi women by utilizing a Bayesian geo-additive regression method. Using the Bayesian approach, this first comprehensive study provides constructive evidence by considering the effects of independent variables for underweight, overweight, or obesity. Notably, as the findings suggested, demographic determinants such as place of residence, wealth, education attainment, working statuses, as well as access to newspapers and television watching behaviors were associated with underweight, overweight/obesity status of Bangladeshi women.

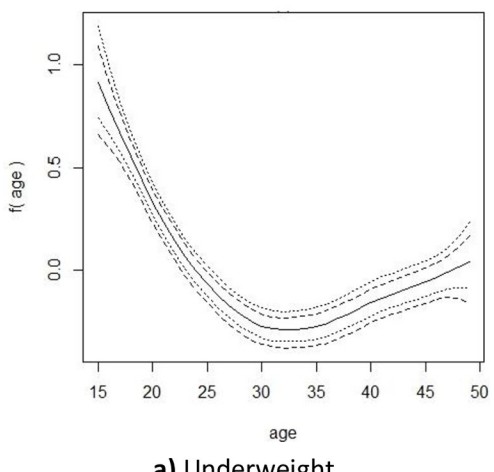

**a)** Underweight

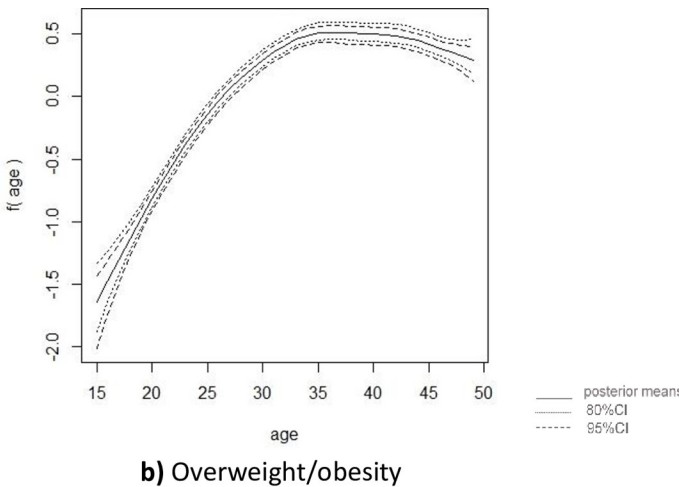

**b)** Overweight/obesity

**Fig 4.** Nonlinear effects of respondent's age for **a)** underweight, **b)** overweight/obesity among reproductive age-group women in Bangladesh (BDHS 2014).

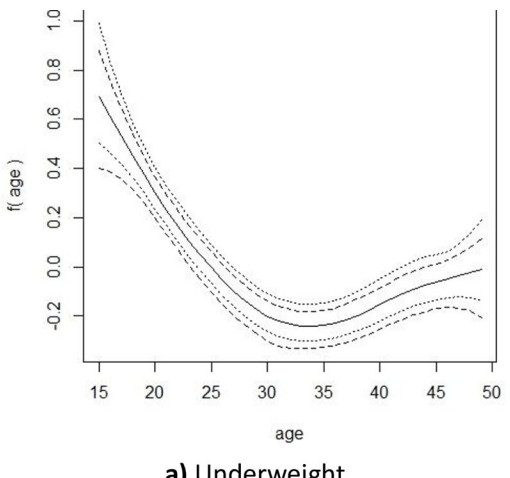
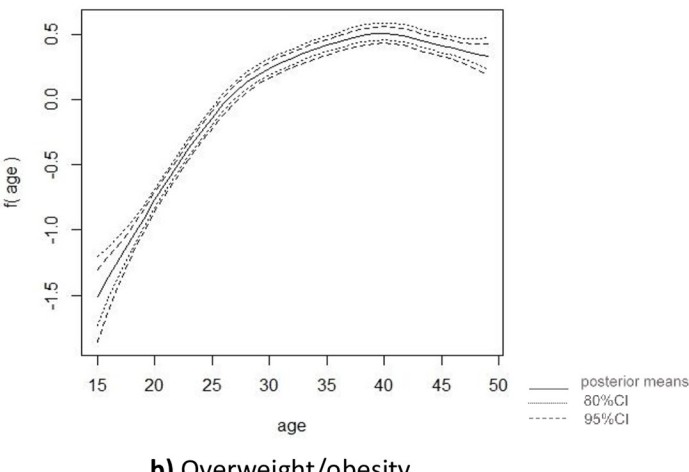

**a)** Underweight

**b)** Overweight/obesity

**Fig 5.** Nonlinear effects of respondent's age for **a)** underweight, **b)** overweight/obesity reproductive age-group women in Bangladesh (BDHS 2017–18).

The findings of our study are consistent with previous works [30, 31]. From 2014 to 2017–18, the prevalence of underweight among reproductive-age Bangladeshi women decreased, nevertheless, that of overweight/obesity increased significantly, reaching more than 50%. Other studies have suggested that overweight/obesity rates in Bangladesh have quadrupled from 1996 to 2011. Since 1996, there has been an increase in the prevalence of overweight and a decrease in the prevalence of underweight, particularly among women [32, 33]. Likewise, the trend of under-nutrition dropped from 32.22% in 2004 to 18.29% in 2014 [31]. Therefore, Bangladesh and other South Asian developing countries (in the same line) have been experiencing a double burden of under-and over-nutrition, prompting a regional rethinking of regional nutrition policy [34, 35].

As per our findings for 2014 survey, women in metropolitan are more likely to be over-weight/obese. The rapid increase in urban population is one of the potential drivers of unequal distributions of nutritional status, particularly urban-centered overweight [36]. In Bangladesh, the percentage of the population living in urban areas has risen over time [37]. The rise in urban population confirms an increase in overweight/obesity in metropolitan and other urbanized areas [38]. Several studies published have repeatedly reported that the rates have risen exponentially, far exceeding the national average (34 percent versus 24 percent) [39]. Moreover, residents in the urban and peripheral areas have easy access to processed food enriched in sugar and unsaturated fat. Globalization, vast investment in food industries, and thriving supermarkets selling energy-dense and junk food at affordable prices have resulted in a shift in food habits and increased unusual bodyweight [40, 41]. Also, the busy work-life in urban avert people to prepare healthy and home-prepared food. Ultimately, they are more likely to consume fast food that is readily accessible. Unhealthy food habits and sedentary behaviors with no or minimal physical activities may be the key contributors to body mass and the development of non-communicable diseases [42]. A study from Bangladesh reported high rates of overweight/obesity in urban areas compared with rural and increased prevalence of diabetes and hypertension [43]. Evidence-based preventive measures and a strong political commitment would help curb the rising prevalence of overweight/obesity. A slow progress in addressing under-nutrition among urban poor and rural populations, as well as widespread urban over-nutrition, have added a systemic burden to the healthcare delivery system in low-middle income countries.

The co-occurrence of under- and over-weight people in a country is a concerning situation that necessitates location-specific nutrition intervention programs to address the nutrition issues. Our study findings are consistent with the figures generated by using Asian body mass index cut-offs, which reported co-prevalence of under and over-nutrition in neighboring countries such as India [44], Myanmar [45], Nepal [35], and Pakistan [46].

Our study aimed to address the division-specific likelihood of women for being underweight and overweight. Results indicate that women from Sylhet were more likely to be underweight for both surveys. Overweight/obesity was common among women in Rajshahi and Khulna; however, the likelihood of women being overweight/obese shifted to Khulna, Barishal, and Chittagong in 2017–18. The findings are staggering as changing status of overweight/obesity depicts a systemic inability to address the nutrition needs of the region. Therefore, it is essential to scrutinize the regional variation in women's body mass index and identify the local level factors contributing to the unrestrained/high prevalence. Accordingly, imperative actions could be taken for quick improvement in the nutritional status of women across regions.

In line with the literature, our study found that women from lower-income households were more likely to be underweight, while women from higher-income households were more likely to be overweight or obese [17, 32, 47]. Despite the fact that the wealthy population has better housing than the poor, the wealthy population's lifestyle and eating habits have been drastically changed. To some extent, the richest quintile households tend to consume more calories than they require. Another obvious explanation is that wealthier women buy food from supermarkets and eat out frequently. At the same time, richer households are exposed to readily available cheap, fast food or street food often prepared by adding extra amount of oil and ghee [48]. Regular consumption of such foods contributes to an increase in the risk of being overweight/obese. Contrarily, poor households have limited purchasing power to buy enough food to meet their daily needs. They prefer to eat food prepared at home. Consequently, they are less likely to consume a high-calorie diet. This finding demonstrates the importance of carefully assessing the wealth status of target communities or localities when developing nutrition programs.

Our study also found differences in the likelihood of being underweight and overweight among women with low and higher education. Having primary or secondary education was linked to less likelihood of being underweight and more likelihood of being overweight/obese. Contrarily, highly educated women were less likely to be overweight/obese. In general, education enables individuals to understand the importance of healthy and nutritious intake and physical fitness. Nevertheless, studies from South Asian countries reported ambiguous results regarding education and women's body mass index. Studies published earlier showed that women with post-secondary education were less likely to be overweight and obese [34, 49].

As per our study, working women were less likely to be overweight/obese. It would be worth noting that more than 60% of the workforce in Bangladesh are in low-paid jobs, such as laborer, factory workers, domestic and manufacturing workers. Mostly, these working women are poor and cannot afford to include adequate nutrition in their everyday meals. Likewise, women working in these low-profile jobs are engaged in intensive physical work, therefore, they have lower chances of being overweight/obese. Access to mass media is vital for receiving information on nutritious diet and healthy practices. Our study findings showed significant association between nutrition outcomes among Bangladeshi women with access to newspapers and television. Reading newspapers at least once a week and watching television at least once a week and/or almost every day was associated with a higher likelihood of being overweight/obese. The role of mass media in acquiring nutrition-related information is undeniable. However, there are limited studies that have assessed the correlation between mass media and nutritional status. Studies from Bangladesh [17, 50] and Ethiopia [51, 52] and Nepal [35]

reported positive association between television watching and overweight or obesity. Unlike other forms of mass media, television watching is considered a luxury activity and usually spends time in a sitting position. This immobile sedentary activity is also linked with the consumption of unhealthy eating [53], usually the intake of high caloric and fast food [54]. Besides, commercial advertisements are also one of the influencing factors for purchasing junk foods [55].

This study has a few limitations. Firstly, the analysis of this study is limited to reproductive-age women. Therefore, it cannot be generalized to women beyond this age-group. Secondly, the important individual-level characteristics such as food diversity, diet intake and physical activity which affect women's body mass could not be included in the analysis, since they were not collected in BDHS surveys; Thirdly, the spatial data collected in the BDHS 2014 was different from that of BDHS 2017–18. Mymensingh was separated from Dhaka Division on 14 September 2015 [56], which limited us to determine BMI status of women according to the new division of the country for 2017–18 survey. Further, due to the inability to deploy district-level geographical maps into the analysis, in-depth geographic variation of under and overweight could not be assessed.

In Bangladesh, women faced a double burden of underweight and overweight/obesity, with a significant increase in the prevalence of overweight/obesity in 2017–18 compared to 2014. In Sylhet, women were more likely to be underweight, whereas in Khulna, overweight/obesity was consistently significant in two consecutive surveys. The unabated rise in overweight/obesity among urban women necessitates a radical shift in nutrition interventions to combat its rising rates in highly urbanized cities. Similarly, broader population-level intervention through mass media could be possible to promote eating a nutritious diet, behavior change, and highlight sedentary activities such as television.

## Acknowledgments

Data for this study was provided by the Measure DHS and we appreciate Measure DHS for granting access to the data and boundary file. The data used in this study can be accessed by registering at https://dhsprogram.com/Data/.

## Author Contributions

**Conceptualization:** Richa Vatsa, Umesh Ghimire.

**Data curation:** Richa Vatsa, Umesh Ghimire.

**Formal analysis:** Richa Vatsa, Umesh Ghimire.

**Investigation:** Khaleda Yasmin, Farhana Jesmine Hasan.

**Methodology:** Richa Vatsa.

**Supervision:** Khaleda Yasmin, Farhana Jesmine Hasan.

**Visualization:** Richa Vatsa, Umesh Ghimire.

**Writing – original draft:** Umesh Ghimire.

**Writing – review & editing:** Richa Vatsa, Khaleda Yasmin, Farhana Jesmine Hasan.

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
