## [Decision Letter · Decision Letter 0]

4 Jul 2022

PONE-D-21-26719Modelling spatial variation and determinants of under and over nutrition among reproductive-age women in BangladeshPLOS ONE

Dear Dr. Ghimire,

Thank you for submitting your manuscript to PLOS ONE. After careful consideration, we feel that it has merit but does not fully meet PLOS ONE’s publication criteria as it currently stands. Therefore, we invite you to submit a revised version of the manuscript that addresses the points raised during the review process.

Specifically, we require experiments, statistics, and other analyses are performed to a high technical standard and are described in sufficient detail. The reviewers raised a couple of concerns about your methodology including sample size and statistical analysis. Please have all the comments addressed point-by-point.

We look forward to receiving your revised manuscript.

Kind regards,

Jianhong Zhou

Staff Editor

PLOS ONE

Journal Requirements:

a) Did participants provide their written or verbal informed consent to participate in this study?

3. Thank you for stating the following financial disclosure: "The authors did not receive any grants or financial support for this study, neither it was prepared as a part of the employment or academic studies."

5. Please upload a copy of Figures 1-3, to which you refer in your text on page 18. If the figure is no longer to be included as part of the submission please remove all reference to it within the text.

6. We note that Figure S1 in your submission contain map images which may be copyrighted. All PLOS content is published under the Creative Commons Attribution License (CC BY 4.0), which means that the manuscript, images, and Supporting Information files will be freely available online, and any third party is permitted to access, download, copy, distribute, and use these materials in any way, even commercially, with proper attribution. For these reasons, we cannot publish previously copyrighted maps or satellite images created using proprietary data, such as Google software (Google Maps, Street View, and Earth). For more information, see our copyright guidelines: http://journals.plos.org/plosone/s/licenses-and-copyright.

a. You may seek permission from the original copyright holder of Figure S1 to publish the content specifically under the CC BY 4.0 license.  

Reviewers' comments:

Reviewer's Responses to Questions

**Comments to the Author**

1. Is the manuscript technically sound, and do the data support the conclusions?

Reviewer #1: Yes

Reviewer #2: Partly

Reviewer #3: Yes

2. Has the statistical analysis been performed appropriately and rigorously? 

Reviewer #1: Yes

Reviewer #2: No

Reviewer #3: Yes

3. Have the authors made all data underlying the findings in their manuscript fully available?

Reviewer #1: Yes

Reviewer #2: Yes

Reviewer #3: Yes

4. Is the manuscript presented in an intelligible fashion and written in standard English?

Reviewer #1: Yes

Reviewer #2: No

Reviewer #3: Yes

5. Review Comments to the Author

Reviewer #1: I congratulate the authors for producing a quality of research work based on national level representative data of Bangladesh. There are studies based on the conclusion of the present study as the problem of double burden of malnutrition in Bangladesh but the approach is different in the present study with rigorous and appropriate statistical analysis. Only I suggest to look into the English language, meticulous editing and some of the more references in Asian context and Bangladesh in particular.

Reviewer #2: Abstract:

The sample size needs to be mentioned under methods

Under results, need to include values of statistics e.g coeffecients and thier CI.

under results, last line, thee is a typo i.e ..... in the year 017/18

Introduction:

Write " LMIC"' in full at first then abbreviations later.

Methods:

You are using a multinomial logit, please specify the reference category of the outcome?

How did you take into account design effect of the DHS survey in you analysis? Routine analysis of DHS data usually take into account survey design.

Please mention how you selected your independent variables. What was the selection criteria? Did you just guess?

I am wondering the p-value in Table 1, is it testing for association or comparing proportions? And if its testing for association, between what variables? if its comparing proportions, which ones? Furthermore, is the p-value for chi-square or t-test?. May the authors describe what kind of descriptive analysis was done in Table 1 under statistical analysis section i.e was it chi-square or t-test?

Results:

DIC is not necessary as you did not fit other models to compare with in table 2. To maintain DIC, I would love if the authors first fit linear model with an all linear terms and then the one with non-linear terms with spatial and then compare using the DIC. This would validate if the use of non-linear and spatial terms is necessary or not.

In table 2, include the variance parameters for non linear and spatial effects. Interpret the variance parameters.

Footer of table 2 "*Significant as mean value is included in the 95% credible interval" must be "*Significant as zero is not included in the 95% credible interval"

In abstract under conclusion section authors are saying ...Underweight and overweight are uneven across the

country....,The authors were also supposed to test for randomness in spatial distribution using Moran I test or any other related test.

Discussion:

Under weaknesses, the authors are mentioning that there is no district level maps. Is this true? Every country has district level maps. Bangladesh is not a new country. The authors must be honest. This analysis would very insightful if was done at smaller spatial scale than done in this study at larger regional/provincial level.

Grammar

Authors need to proof read the text, there are a lot of typos and poor sentence construction and punctuations.

References:

Please follow the journal guidelines in writing references. Some references are incomplete without page ranges e.g reference 52 and 53. I suggest authors to proof read all references.

Reviewer #3: This is an important area of research in Bangladesh with good sample size to support the study findings. There are important concerns with that needs to be address, however, that require major revision.

6. PLOS authors have the option to publish the peer review history of their article (what does this mean?). If published, this will include your full peer review and any attached files.

Reviewer #1: No

Reviewer #2: No

Reviewer #3: No

---

## [Author Response · Author response to Decision Letter 0]

30 Sep 2022

Dear Editor,

Thank you for giving us an opportunity to revise our manuscript. This paper is an important in terms of statistical methods and approach that we employ and provide a robust result on nutritional status of women. 

Here are our responses to the reviewer’s comments.

This is an important area of research in Bangladesh with good sample size to support the study findings. There are important concerns with that needs to be address, however, that require major revision.

Authors may consider following suggestions for revising the manuscript:

Comment: The missing of page number and line number becomes difficult for the reviewers to mention the place for making the comments and suggestion.

Author’s reply: We have added the page and line number in the manuscript. Apologies that it wasn’t there in the initial version.

Comment: It can also be mentioned that why traditional linear regression models may fail to capture the spatial effects and why you choose Bayesian Geo-additive model. Very briefly, it can also be highlighted about the importance of accounting for spatial autocorrelation and how it can lead to bias in estimates.

Author’s reply: Thank you for this comment. We have added two paragraphs to answer the related queries into the methodology section between the line numbers 160 to 165, and 145 to 149. 

Comment: It will be interesting if the authors perform the analysis by splitting the spatial effects into structured and unstructured spatial effects.

Author’s reply: We have performed the analysis by splitting the spatial effects for structured and unstructured spatial effects and have also included the rationale for utilizing spatial effects.

Comment: The authors must perform several model diagnostic tests to arrive at the best model, and incorporate the model selection parameters in the manuscript. 

I would suggest reading the paper by Marbaniang et al., (2022) on model diagnostic and spatial effect.

Author’s reply: Thank you for suggesting the paper, which is worth reading. We have used the DIC, deviance values, for each of the six models fitted for both surveys- 2014 and 2017-18. The results of DIC suggested that models M4, M5, and M6 have the lowest DICs (<5) for both surveys indicating that any of the three models can be considered equally good for model fitting of the data. We have selected model M4 for the estimation of effects.

Comment: It is advisable to use two more models here to understand the influence of spatially structure and spatially unstructured heterogeneity. In this way it can be known about the drop/increase in DIC

Author’s reply: To address this comment, we analyzed three different models so as to account for possible separate or joint effects of spatially structured and spatially unstructured heterogeneity in data. However, all these three models have very similar DIC values (with differences less than 1); and thus can be considered equally good to explain variability in data. Therefore, we present results of the model including structured spatial effects, linear, and nonlinear effects only. Accordingly, the methodology and results sections have been modified to address the comment. 

Comment: Instead of using wealth index as a linear effect. A continuous variable “wealth score” can be used to explore the non-linear effect.

Author’s reply: Based on the available literature that used the DHS data, we have concluded that we should stick with the categorized wealth index to explore its linear effect.

Comment: You need to mention how you are defining your neighbor here? Did you use adjacency metric or any other method to define neighborhood, then you need to mention about it and may write the equation/formula for the same

Author’s reply: To address this comment, we have added a couple of sentences defining neighbouring regions under methodology section between line numbers 180 to 182. 

Reviewer #1: I congratulate the authors for producing a quality of research work based on national level representative data of Bangladesh. There are studies based on the conclusion of the present study as the problem of double burden of malnutrition in Bangladesh but the approach is different in the present study with rigorous and appropriate statistical analysis. Only I suggest to look into the English language, meticulous editing and some of the more references in Asian context and Bangladesh in particular.

Reviewer #2: Abstract:

The sample size needs to be mentioned under methods

Author’s reply: The information is already mentioned in the Method section.

Under results, need to include values of statistics e.g coeffecients and thier CI.

Author’s reply: Appropriate statistical estimates (posterior mean) and their 95% credible interval are presented in the table.

under results, last line, thee is a typo i.e ..... in the year 017/18

Author’s reply: The typo has been corrected. 

Introduction:

Write " LMIC"' in full at first then abbreviations later.

Author’s reply: This issue has been addressed. 

Methods:

You are using a multinomial logit, please specify the reference category of the outcome?

Author’s reply: We have added the reference categories to (now) Table 3 showing the results for the linear effects. 

How did you take into account design effect of the DHS survey in you analysis? Routine analysis of DHS data usually takes into account survey design. 

Author’s reply: Thank you for pointing out this query. All the analysis were performed so that it addresses the complex sampling design, non-response rate and oversampling of population in the survey. 

Please mention how you selected your independent variables. What was the selection criteria? Did you just guess?

Author’s reply: The selection of independent variables was based on the literature reviews from different sources. Literatures were discussed in the introduction and discussion section of the article.

I am wondering the p-value in Table 1, is it testing for association or comparing proportions? And if its testing for association, between what variables? if its comparing proportions, which ones? Furthermore, is the p-value for chi-square or t-test?. May the authors describe what kind of descriptive analysis was done in Table 1 under statistical analysis section i.e was it chi-square or t-test?

Author’s reply: The pvalue in the table one was calculated from the t-test for proportion to compare the proportion change between two survey periods. We have added a note indicating this below Table 1 and have also added a sentence regarding it into the Results section. 

Results:

DIC is not necessary as you did not fit other models to compare with in table 2. To maintain DIC, I would love if the authors first fit linear model with an all linear terms and then the one with non-linear terms with spatial and then compare using the DIC. This would validate if the use of non-linear and spatial terms is necessary or not.

Author’s reply: Thank you for the suggestion to include different models to incorporate linear, nonlinear, and spatial effects, separately or jointly; we have followed it into the study and fitted six different models. We have added a separate table to compare the models through DIC values. Upon comparison, we analyzed and presented the results for the model with the least DIC. 

Anything that you can write in repose to the above comments

In table 2, include the variance parameters for non linear and spatial effects. Interpret the variance parameters.

Author’s reply: Estimates of variance parameters have been added to (now) Table 3 and corresponding texts have been also added to the Results section under headings-Spatial Effects and Non-linear Effects, respectively. 

Anything that you can write in repose to the above comments

Footer of table 2 "*Significant as mean value is included in the 95% credible interval" must be "*Significant as zero is not included in the 95% credible interval"

Author’s reply: We have added the footer so that it is clear to read the table. 

In abstract under conclusion section authors are saying ...Underweight and overweight are uneven across the country....,The authors were also supposed to test for randomness in spatial distribution using Moran I test or any other related test.

Author’s reply: 

If this is not addressed/not possible to address just mention a sentence in the limitation section

Author’s reply: We have modified the sentence to reflect the results of this study which utilizes GMRF priors to model effects of spatial autocorrelation. 

Discussion:

Under weaknesses, the authors are mentioning that there is no district level maps. Is this true? Every country has district level maps. Bangladesh is not a new country. The authors must be honest. This analysis would very insightful if was done at smaller spatial scale than done in this study at larger regional/provincial level.

Author’s reply: This was one of our concerns when conceptualizing the analysis of manuscript. However, DHS provides the shape files up to the division levels of Bangladesh. The reason that we abort this idea is driven by the assumption that the way BDHS samples are selected using cluster sampling technique which might not provide the accurate estimate for the lower administrative levels.

Grammar

Authors need to proof read the text, there are a lot of typos and poor sentence construction and punctuations.

Author’s reply: We have revised the article and proof read by the native English language speaker.

References:

Please follow the journal guidelines in writing references. Some references are incomplete without page ranges e.g reference 52 and 53. I suggest authors to proof read all references.

Author’s reply: We have individually revised the references.

---

## [Decision Letter · Decision Letter 1]

31 Oct 2022

PONE-D-21-26719R1Determinants of under- and over- nutrition among reproductive-age women in Bangladesh: trend analysis using spatial modeling

PLOS ONE

Dear Dr. Ghimire,

Thank you for submitting your manuscript to PLOS ONE. After careful consideration, we feel that it has merit but does not fully meet PLOS ONE’s publication criteria as it currently stands. Therefore, we invite you to submit a revised version of the manuscript that addresses the points raised during the review process.

After receiving the comments form the reviewers I would invite you to do minor revision as per the suggestion of Reviewer 2. The comment can be found below in Section 6

We look forward to receiving your revised manuscript.

Kind regards,

Strong P Marbaniang

Guest Editor

PLOS ONE

Journal Requirements:

Additional Editor Comments:

The paper is in good shape now after incorporating the comments, however I would suggest some minor revision to the revise manuscript by incorporate the comment suggested by the Reviewer 2.

Reviewers' comments:

REVIEWER#2

I am satisfied with the response of the authors as they have incorporated all the comments and suggestions. However, I would suggest the authors to cite the related paper by Marbaniang et al., (2022) as it is related to Bayesian Technique used in your paper.

**Comments to the Author**

1. If the authors have adequately addressed your comments raised in a previous round of review and you feel that this manuscript is now acceptable for publication, you may indicate that here to bypass the “Comments to the Author” section, enter your conflict of interest statement in the “Confidential to Editor” section, and submit your "Accept" recommendation.

Reviewer #2: All comments have been addressed

Reviewer #3: All comments have been addressed

2. Is the manuscript technically sound, and do the data support the conclusions?

Reviewer #2: Yes

Reviewer #3: Yes

3. Has the statistical analysis been performed appropriately and rigorously? 

Reviewer #2: Yes

Reviewer #3: Yes

4. Have the authors made all data underlying the findings in their manuscript fully available?

Reviewer #2: No

Reviewer #3: Yes

5. Is the manuscript presented in an intelligible fashion and written in standard English?

Reviewer #2: Yes

Reviewer #3: Yes

6. Review Comments to the Author

Reviewer #2: The authors have adressed most of the coments. Even though some coments on abstract have been neglected. But overall, the paper can be published.

Reviewer #3: I am satisfied with the response of the authors as they have incorporated all the comments and suggestions. However, I would suggest the authors to cite the related paper by Marbaniang et al., (2022) as it is related to Bayesian Technique used in your paper.

I wish the authors all the best and I recommend the paper for publication.

7. PLOS authors have the option to publish the peer review history of their article (what does this mean?). If published, this will include your full peer review and any attached files.

Reviewer #2: **Yes: **Alfred Ngwira

Reviewer #3: No

---

## [Author Response · Author response to Decision Letter 1]

8 Nov 2022

As suggested by the reviewer, we have cited Marbaniang et al., (2022) in the methods section.

---

## [Decision Letter · Decision Letter 2]

20 Jan 2023

PONE-D-21-26719R2Determinants of undernutrition and overnutrition among reproductive-age women in Bangladesh: trend analysis using spatial modelingPLOS ONE

Dear Dr. Ghimire,

Thank you for submitting your manuscript to PLOS ONE. After careful consideration, we feel that it has merit but does not fully meet PLOS ONE’s publication criteria as it currently stands. Therefore, we invite you to submit a revised version of the manuscript that addresses the points raised during the review process.Please take a look the reviewers comments, especially, comments on the introduction and methods sections.

We look forward to receiving your revised manuscript.

Kind regards,

Md Jamal Uddin, Ph.D

Academic Editor

PLOS ONE

Journal Requirements:

Reviewers' comments:

Reviewer's Responses to Questions

**Comments to the Author**

1. If the authors have adequately addressed your comments raised in a previous round of review and you feel that this manuscript is now acceptable for publication, you may indicate that here to bypass the “Comments to the Author” section, enter your conflict of interest statement in the “Confidential to Editor” section, and submit your "Accept" recommendation.

Reviewer #3: All comments have been addressed

Reviewer #4: (No Response)

2. Is the manuscript technically sound, and do the data support the conclusions?

Reviewer #3: Yes

Reviewer #4: Yes

3. Has the statistical analysis been performed appropriately and rigorously? 

Reviewer #3: Yes

Reviewer #4: Yes

4. Have the authors made all data underlying the findings in their manuscript fully available?

Reviewer #3: Yes

Reviewer #4: Yes

5. Is the manuscript presented in an intelligible fashion and written in standard English?

Reviewer #3: Yes

Reviewer #4: Yes

6. Review Comments to the Author

Reviewer #3: I am satisfied with the response of the authors and I have no further comments. Hence I recommend the paper for publication.

Reviewer #4: Introduction: The introduction section's first two paragraphs (number lines 52–64) were poorly written. The sentence are not interlinked with previous one . For instance: The authors highlighted the enormous difficulties and worldwide complexity of undernutrition in the first sentence (line 52), yet they discuss malnutrition in the next two lines. Malnutrition includes undernutrition, according to the WHO. These two things cannot be mixed. It would be better to concentrate solely on it or provide clarification first because the authors used undernutrition in the title.

Methodology: The authors did not specify whether they used sample weight or not while analyzing the complex survey data from the BDHS. While analyzing complex survey data, like those from BDHS, MICS, etc., sampling weight must be taken into consideration.

7. PLOS authors have the option to publish the peer review history of their article (what does this mean?). If published, this will include your full peer review and any attached files.

Reviewer #3: No

Reviewer #4: **Yes: **Md Sabbir Hossain

While revising your submission, please upload your figure files to the Preflight Analysis and Conversion Engine (PACE) digital diagnostic tool, https://pacev2.apexcovantage.com/. PACE helps ensure that figures meet PLOS requirements. To use PACE, you must first register as a user. Registration is free. Then, login and navigate to the UPLOAD tab, where you will find detailed instructions on how to use the tool. If you encounter any issues or have any questions when using PACE, please email PLOS at figures@plos.org. Please note that Supporting Information files do not need this step.<quillbot-extension-portal></quillbot-extension-portal>

---

## [Author Response · Author response to Decision Letter 2]

12 Feb 2023

Dear Editor,

Thank you again for giving us an opportunity to revise our manuscript. The feedback provided by reviewers are important in terms of statistical methods and approach that we employ and provide a robust result on nutritional status of women. 

Here are our responses to the reviewer’s comments.

Reviewer's Responses to Questions

Comments to the Author

1. If the authors have adequately addressed your comments raised in a previous round of review and you feel that this manuscript is now acceptable for publication, you may indicate that here to bypass the “Comments to the Author” section, enter your conflict of interest statement in the “Confidential to Editor” section, and submit your "Accept" recommendation.

Reviewer #3: All comments have been addressed

Reviewer #4: (No Response) 

Author’s Response: Thank you.

2. Is the manuscript technically sound, and do the data support the conclusions?

Reviewer #3: Yes

Reviewer #4: Yes

Author’s Response: Thank you.

3. Has the statistical analysis been performed appropriately and rigorously?

Reviewer #3: Yes

Reviewer #4: Yes

Author’s Response: Thank you.

4. Have the authors made all data underlying the findings in their manuscript fully available?

The PLOS Data policy requires authors to make all data underlying the findings described in their As this study utilized the data from the openly accessibl manuscript fully available without restriction, with rare exception (please refer to the Data Availability Statement in the manuscript PDF file). The data should be provided as part of the manuscript or its supporting information, or deposited to a public repository. For example, in addition to summary statistics, the data points behind means, medians and variance measures should be available. If there are restrictions on publicly sharing data—e.g. participant privacy or use of data from a third party—those must be specified.

Reviewer #3: Yes

Reviewer #4: Yes

Author’s Response: Thank you for your concern. We are determined to adhere to the PLOS Data Policy. As we have informed in the Acknowledgement section of this article that the study utilizes open-access data from https://dhsprogram.com. Public use data is accessible after providing justification for the rational use of data. 

5. Is the manuscript presented in an intelligible fashion and written in standard English?

Reviewer #3: Yes

Reviewer #4: Yes

Author’s Response: Thank you for your comments. We have revised our manuscript’s language again, and a Native English Speaker has revised it.

6. Review Comments to the Author

Reviewer #3: I am satisfied with the response of the authors and I have no further comments. Hence I recommend the paper for publication.

Reviewer #4: Introduction: The introduction section's first two paragraphs (number lines 52–64) were poorly written. The sentence are not interlinked with previous one . For instance: The authors highlighted the enormous difficulties and worldwide complexity of undernutrition in the first sentence (line 52), yet they discuss malnutrition in the next two lines. Malnutrition includes undernutrition, according to the WHO. These two things cannot be mixed. It would be better to concentrate solely on it or provide clarification first because the authors used undernutrition in the title.

Author’s Response: The sentences in the introduction are revised and we resolved any misunderstanding. 

Methodology: The authors did not specify whether they used sample weight or not while analyzing the complex survey data from the BDHS. While analyzing complex survey data, like those from BDHS, MICS, etc., sampling weight must be taken into consideration.

Author’s Response: Thank you for raising concern regarding applying sampling weights in the analysis. We would like to emphasize that the study utilizes a Bayesian hierarchical model, which includes a random effect component accounting for the hierarchical nature of the data. So, the unweighted results from the Bayesian analysis are considered valid.

7. PLOS authors have the option to publish the peer review history of their article (what does this mean?). If published, this will include your full peer review and any attached files.

Do you want your identity to be public for this peer review? For information about this choice, including consent withdrawal, please see our Privacy Policy.

Reviewer #3: No

Reviewer #4: Yes: Md Sabbir Hossain

Author’s Response: Thank you.

---

## [Editor Report · Decision Letter 3]

1 Mar 2023

Determinants of undernutrition and overnutrition among reproductive-age women in Bangladesh: trend analysis using spatial modeling

PONE-D-21-26719R3

Dear Dr. Ghimire,

We’re pleased to inform you that your manuscript has been judged scientifically suitable for publication and will be formally accepted for publication once it meets all outstanding technical requirements.

Kind regards,

Md Jamal Uddin, Ph.D

Academic Editor

PLOS ONE
---

## [Editor Report · Acceptance letter]

29 Mar 2023

PONE-D-21-26719R3 

Determinants of undernutrition and overnutrition among reproductive-age women in Bangladesh: trend analysis using spatial modeling 

Dear Dr. Ghimire:

I'm pleased to inform you that your manuscript has been deemed suitable for publication in PLOS ONE. Congratulations! Your manuscript is now with our production department. 

Kind regards, 

on behalf of

Dr. Md Jamal Uddin 

Academic Editor

PLOS ONE